# Light-driven formation of manganese oxide by today's photosystem II supports evolutionarily ancient manganese-oxidizing photosynthesis

Petko Chernev[1,3], Sophie Fischer[1], Jutta Hoffmann[1], Nicholas Oliver[1], Ricardo Assunção[1], Boram Yu[1], Robert L. Burnap [2], Ivelina Zaharieva [1], Dennis J. Nürnberg [1], Michael Haumann[1] & Holger Dau [1✉]

Water oxidation and concomitant dioxygen formation by the manganese-calcium cluster of oxygenic photosynthesis has shaped the biosphere, atmosphere, and geosphere. It has been hypothesized that at an early stage of evolution, before photosynthetic water oxidation became prominent, light-driven formation of manganese oxides from dissolved Mn(2+) ions may have played a key role in bioenergetics and possibly facilitated early geological manganese deposits. Here we report the biochemical evidence for the ability of photosystems to form extended manganese oxide particles. The photochemical redox processes in spinach photosystem-II particles devoid of the manganese-calcium cluster are tracked by visible-light and X-ray spectroscopy. Oxidation of dissolved manganese ions results in high-valent Mn(III, IV)-oxide nanoparticles of the birnessite type bound to photosystem II, with 50-100 manganese ions per photosystem. Having shown that even today's photosystem II can form birnessite-type oxide particles efficiently, we propose an evolutionary scenario, which involves manganese-oxide production by ancestral photosystems, later followed by downsizing of protein-bound manganese-oxide nanoparticles to finally yield today's catalyst of photosynthetic water oxidation.

---

[1] Physics Department, Freie Universität Berlin, Arnimallee 14, 14195 Berlin, Germany. [2] Department of Microbiology and Molecular Genetics, Oklahoma State University, Stillwater, OK 74078-4034, USA. [3] Present address: Department of Chemistry - Ångström Laboratory, Molecular Biomimetics, Uppsala University, Lägerhyddsvägen 1, 75120 Uppsala, Sweden. ✉email: holger.dau@fu-berlin.de

Nature´s invention of photosynthetic water oxidation about three billion years ago (or even earlier[1]) was a breakpoint in Earth´s history because it changed the previously anoxic atmosphere to today´s composition with ~21% $O_2$, practically depleting the oceans of ferrous iron and divalent manganese due to metal-oxide precipitation[2,3]. Water oxidation is catalyzed by a unique bioinorganic cofactor, denoted $Mn_4CaO_5$ according to its oxo-bridged metal core, which is bound to amino acids of the proteins of photosystem II (PSII) in the thylakoid membrane (Fig. 1)[4–8]. This catalyst originally developed in (prokaryotic) cyanobacteria, which were later incorporated by endosymbiosis into the ancestor of the (eukaryotic) cells of algae and plants to yield the chloroplast organelles[9]. The central PSII proteins as well as $Mn_4CaO_5$ (and its main catalytic performance features) are strictly conserved among photosynthetic organisms[10,11].

Native PSII operates as a light-driven oxidoreductase (Fig. 1). Upon sequential excitation with four visible-light photons, four electrons from two bound water molecules are transferred from $Mn_4CaO_5$ to a redox-active tyrosine ($Y_Z$) at the donor side and then via a cofactor chain to terminal quinone acceptors at the stromal side so that two reduced quinols as well as $O_2$ and four protons are released during each catalytic water oxidation cycle[5,7,12–14]. Starting from a $Mn(III)_3Mn(IV)Y_Z$ state, the catalytic cycle involves alternate electron and proton abstraction to reach a $Mn(IV)_4Y_Z^{ox}$ state followed by (concomitant) Mn reduction, O–O bond formation and $O_2$ release (Fig. 1)[15]. The exceptionally efficient $Mn_4CaO_5$ catalyst has inspired development of synthetic water-oxidizing materials[16–18]. Among the wealth of findings on water oxidation by Mn-based catalysts, here the following two results are of particular importance: (i) Self-assembly of the $Mn(III/IV)_4CaO_5$ core in PSII is a light-driven process, involving step-wise oxidation of four solvent $Mn^{2+}$ ions by $Y_Z^{ox}$ coupled to electron transfer to the quinones[19,20]. (ii) Many amorphous Mn oxides of the birnessite type show significant water oxidation activity and share structural as well as functional features with the $Mn_4CaO_5$ core of the biological catalyst[21–25].

The evolutionary route towards the present water oxidation catalyst in PSII is much debated[2,26–31]. It has been hypothesized that before the evolution of oxygenic photosynthesis an ancestral photosystem developed the capability for light-driven oxidation of dissolved $Mn^{2+}$ ions towards the Mn(III/IV) level, thereby providing the reducing equivalents (electrons) needed for primary biomass formation by $CO_2$ fixation[32,33]. Aside from the implications for biological evolution, photosynthetic Mn-oxide formation has significance in the context of recent hypotheses to account for geologic Mn deposits, for example from the early Paleoproterozoic in South Africa[2,33]. Notably, the process of continuous $Mn^{2+}$ oxidation is chemically not trivial, because suitable redox potentials alone are insufficient. Because solitary Mn(III/IV) ions are not stable in aqueous solution, the ability of the photosystem to stabilize high-valent Mn ions by the efficient formation of extended Mn (oxide) structures is pivotal.

Here, the experimental evidence is provided that today's PSII, depleted of its native $Mn_4CaO_5$ complex and the membrane-extrinsic polypeptides, can form a Mn(III/IV) oxide of the birnessite type. By optical (UV–visible) and X-ray absorption spectroscopy, we show that the light-driven oxidation of $Mn^{2+}$ ions results in Mn-oxide nanoparticles which are bound to the photosystem, thereby supporting that also ancient photosystems could have produced Mn oxides and suggesting a viable evolutionary route to today's catalyst of photosynthetic water oxidation.

## Results

### Spinach photosystems depleted of $Mn_4CaO_5$ and extrinsic polypeptides.
Figure 1 shows the arrangement of protein subunits and cofactors in PSII. A recent crystallographic study has revealed that the metal-binding amino acids are similarly arranged in PSII with or without $Mn_4CaO_5$, with the voids in the Mn-depleted photosystem being filled by water molecules[34]. Only in the absence of the Mn-stabilizing extrinsic proteins, sufficient room for the incorporation of a Mn-oxide nanoparticle into the PSII structure may exist (Fig. 1). Therefore, we explored the ability of purified PSII, depleted of $Mn_4CaO_5$ and the extrinsic proteins, to form Mn-oxide species in vitro. PSII-enriched membrane particles were prepared from spinach[35] and Mn depletion was achieved using an established protocol (see Supplementary Information)[36]. The resulting PSII preparation was inactive in light-driven $O_2$-evolution and Mn was practically undetectable, i.e., <0.2 Mn ions per PSII were found (Table 1). Concomitantly with Mn depletion, the three proteins bound to

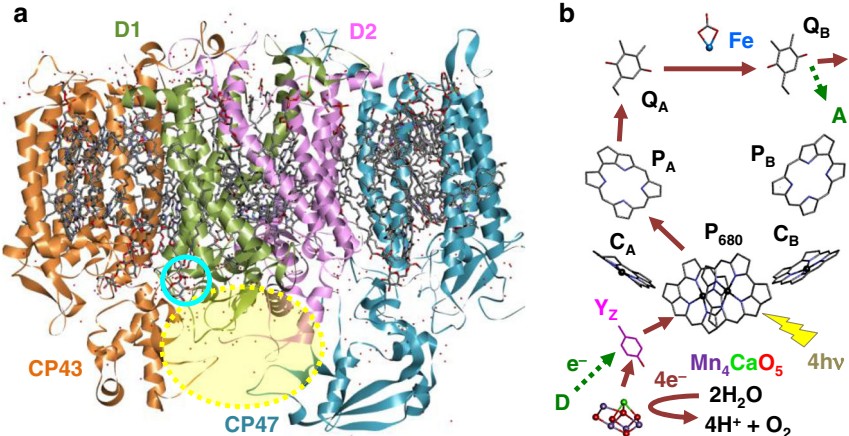

**Fig. 1 Structure and function of photosystem II. a** Cryo-electron microscopy structure of native plant PSII (PDB entry 5XNL). The central membrane-integral subunits with cofactors of a PSII monomer are shown (cyan circle = $Mn_4CaO_5$ position). Omitting the extrinsic subunits (PsbQ, PsbP, and PsbO with approximate molecular weights of 18, 24, and 33 kDa) exposes a cavity (yellow shading, the subunits are absent in our Mn-depleted PSII). **b** Cofactors (truncated structures) for light-driven electron transfer in D1/D2 ($Y_Z$, tyrosine electron acceptor from $Mn_4CaO_5$; $P_{680}$, primary donor chlorophyll (chl) dimer; $C_{A,B}$, accessory chl; $P_{A,B}$, pheophytins; $Q_{A,B}$, acceptor quinones; Fe, non-heme iron with bicarbonate ligand). Green/dark-red arrows: electron transfer paths upon photo-excitation ($h\nu$) of $O_2$-evolving or Mn-depleted PSII (D/A, external electron donor/acceptor).

**Table 1 Metal content of PSII preparations (for details, see Supplementary Table 1).**

| Preparation | Mn per PSII | Fe per PSII |
|---|---|---|
| Intact ($O_2$ evolving) PSII | 4 ± 1 | 9 ± 2 |
| Mn-depleted (non-$O_2$ evolving) PSII | <0.2 ± 0.2 | 6 ± 2 |
| Mn-depleted PSII + 250 µM $MnCl_2$, dark | 7 ± 2 | 5 ± 1 |
| Mn-depleted PSII + 250 µM $MnCl_2$, light | 65 ± 19 | 5 ± 1 |

**Fig. 2 Redox reaction associated with a color change of DCPIP (2,6-dichlorophenol-indophenol).** The color of the oxidized molecule (DCPIP$^{ox}$, left) is blue; the reduced molecule (DCPIP$^{red}$, right) is colorless.

the lumenal side of plant PSII (the so-called extrinsic proteins PsbQ, PsbP, and PsbO)[37,38] were removed, as revealed by polarography, total reflection X-ray fluorescence (TXRF)[39] metal quantification, and gel electrophoresis (Supplementary Figs. 1 and 2; Table 1, Supplementary Table 1).

**UV–vis spectra monitor PSII electron transfer.** Optical absorption spectroscopy was employed for time-resolved tracking of PSII redox chemistry (Figs. 2–4 and Supplementary Figs. 3–8 and 10, 11). DCPIP (2,6-dichlorophenol-indophenol) was added as an artificial electron acceptor for PSII that allows facile optical monitoring of light-driven electron flow[40]. Oxidized DCPIP (DCPIP$^{ox}$) at pH ≥ 7 shows strong absorption at 604 nm and its bright blue color vanishes upon reduction (Figs. 2 and 3, see also Supplementary Fig. 4)[41]. We explored the ability of DCPIP$^{ox}$ to support and simultaneously probe PSII electron transfer by recording absorption spectra as a function of illumination period, light intensity and DCPIP concentration using intact (not Mn-depleted) PSII membrane particles (Supplementary Figs. 3–5). Mn-depleted PSII showed a completely different electron transfer behavior (Figs. 3, 4 and Supplementary Figs. 6–8 and 11). By adding a Mn salt ($MnCl_2$), we investigated hexaquo-$Mn^{2+}$ ions as an exogenous electron donor to PSII. $Mn^{2+}$ in solution is completely colorless, i.e., it does not absorb in the 300–900 nm region. In the dark in the presence of DCPIP$^{ox}$ and $Mn^{2+}$ ions or upon illumination in the absence of $Mn^{2+}$, we did not observe any major spectral change of the PSII suspension (aside from minor bleaching of PSII chlorophyll due to oxidative damage[42]). No absorption changes accountable to DCPIP (i.e., due to its reduction) under illumination were observed in the presence of $MnCl_2$ with simultaneous absence of PSII (Supplementary Fig. 10). These experiments verify for the Mn-depleted PSII: electron transfer towards DCPIP requires both, visible light to drive the PSII electron transfer reactions and $Mn^{2+}$ ions that can serve as a donor in the light-induced electron transfer.

**Electron donation by $Mn^{2+}$ ions.** At low $MnCl_2$ concentration (5 µM), a linear decrease in the amount of DCPIP$^{ox}$ indicates a small and constant rate of electron donation at the PSII donor side (Figs. 3, 4), which is only about 8% of the level reached in intact PSII due to water oxidation (see Supplementary Figs. 7–9 for further data and discussion). This slow electron transfer is not visible at higher $MnCl_2$ concentrations suggesting that at high $MnCl_2$ concentrations, the PSII centers do not maintain a metal

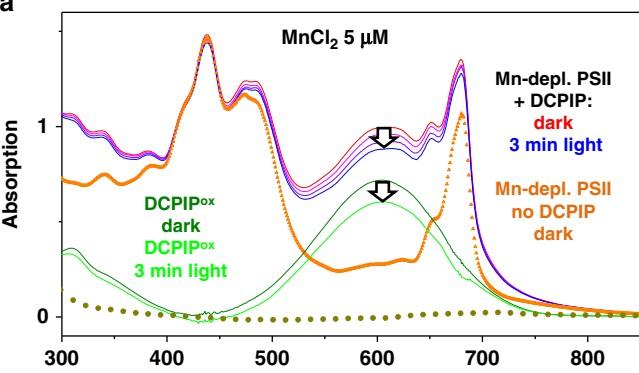

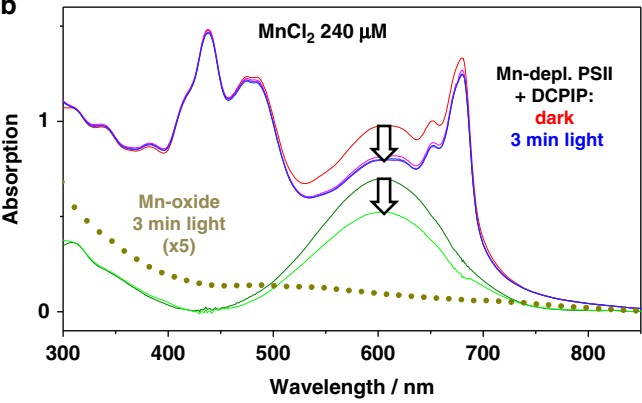

**Fig. 3 Light-driven redox reactions in Mn-depleted PSII tracked by optical spectroscopy. a**, **b** UV–vis absorption spectra. The orange line (triangles) represents a suspension of Mn-depleted PSII before the addition of DCPIP (60 µM) serving as artificial electron acceptor (buffer conditions: 1 M glycine-betaine, 15 mM NaCl, 5 mM $CaCl_2$, 5 mM $MgCl_2$, 25 mM MES buffer, pH 7; for further details see SI). Then DCPIP$^{ox}$ and either 5 µM $MnCl_2$ (in **a**) or 240 µM $MnCl_2$ (in **b**) were added, followed by continuous white-light illumination (1000 µE m$^{-2}$ s$^{-1}$) of the PSII suspensions and collection of spectra (one spectrum per minute; immediately prior to illumination, red lines, and after 3 min light, blue lines). The green and light-green lines correspond to the spectral contributions of DCPIP$^{ox}$ to the dark and 3-min light spectra; the amplitude decrease at 604 nm represents the loss of DCPIP$^{ox}$ due to its reduction by PSII (arrows). The dark-yellow dotted spectra are assignable to a Mn oxide, as verified by X-ray absorption spectroscopy (spectra obtained by weighted spectral deconvolution, see caption of Supplementary Fig. 6, and scaled by a factor of 5, for clarity). Note that significant spectral changes due to Mn-oxide formation were only observed with 240 µM $MnCl_2$ (in **b**), but not with 5 µM $MnCl_2$ (in **a**). Source data are provided as a Source Data file.

site that supports continuous low-rate electron transfer. Therefore we consider this phenomenon, albeit of clear interest for future investigation, irrelevant for the analysis of oxide formation herein observed at higher $MnCl_2$ concentrations. For increasing $MnCl_2$ concentrations, a clearly more rapid phase of DCPIP$^{ox}$ reduction grew in (Fig. 3, arrows; Fig. 4 and Supplementary Fig. 8). Its amplitude saturated at 240 µM $MnCl_2$ and indicates reduction of ~17% (~10 µM) of the initial DCPIP$^{ox}$ (60 µM), which corresponds to up to 200 transferred electrons per PSII within about 1 min. Notably, at low $MnCl_2$ concentration DCPIP reduction continued at undiminished rate for a clearly longer illumination time (>5 min, Fig. 4 and Supplementary Fig. 7), suggesting that photoinhibitory damage does not rapidly terminate electron transfer in the Mn-depleted PSII. Based on the Mn concentration dependence (Fig. 4 and Supplementary Fig. 8) and the results presented in the following, we can assign this rapid DCPIP

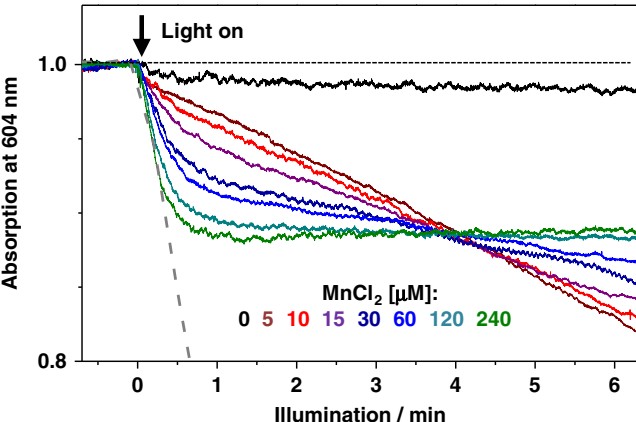

**Fig. 4 Kinetics of light-dependent electron flow in Mn-depleted PSII at various concentrations of solvated Mn$^{2+}$ ions.** The decrease in absorption at 604 nm monitors the light-driven reduction of the artificial electron acceptor (60 μM DCPIP$^{ox}$, pH 7) due to the oxidation of Mn$^{2+}$ ions by PSII. See Supplementary Fig. 7 for traces on a longer time scale. The gray dashed line illustrates the rate of DCPIP reduction by fully intact PSII under similar conditions (Supplementary Fig. 5). Source data are provided as a Source Data file.

reduction phase to oxidation of Mn$^{2+}$ ions and formation of high-valent Mn(III/IV) oxide particles. At 240 μM MnCl$_2$, DCPIP reduction is completed after about 1 min of illumination likely indicating that further electron donation to the PSII donor side (and thus further DCPIP reduction) was impaired after the formation of a Mn-oxide particle at the PSII donor side, possibly by blocking access of further Mn$^{2+}$ ions to the oxidant, which is the tyrosine (Y$_Z$) radical (see Fig. 1).

**UV–vis spectra point towards Mn-oxide formation**. To search for evidence of Mn-oxide formation, informative absorption difference spectra of Mn-depleted PSII before and after illumination were calculated (Fig. 3 and Supplementary Fig. 6). For 60 μM MnCl$_2$, after completion of rapid DCPIP$^{ox}$ reduction (3 min), there was a broad absorption increase (ranging from 350 to 700 nm), which is similar to the wide-range absorption of Mn oxides[25]. For higher concentrations of MnCl$_2$, the absorption assigned to Mn oxides gained strength and became maximal at 240 μM MnCl$_2$ (Fig. 3). Using alternative electron acceptors (Supplementary Fig. 11), similar or even higher Mn-oxide amounts were detected with DCBQ (2,5-dichloro-1,4-benzoquinone) or PPBQ (phenyl-p-benzoquinone), resembling the native quinone acceptor (Q$_B$), but the slow (hydrophilic) acceptor ferricyanide (K$_3$Fe$^{III}$(CN)$_6$) did not yield significant Mn-oxide formation.

**PSII with 50–100 bound Mn ions prepared for analysis by X-ray spectroscopy**. To investigate the Mn$^{2+}$ oxidation products and identify their atomic structure, we employed X-ray absorption spectroscopy (XAS) at the Mn K-edge (Fig. 5 and Supplementary Figs. 12–16). Mn-depleted PSII was illuminated for 3 min with 240 μM MnCl$_2$ and 60 μM PPBQ$^{ox}$ at pH 8.5 or pH 7 (Fig. 5, Supplementary Figs. 14 and 15), the reaction was terminated by rapid sample cooling in the dark, and the PSII membranes were pelleted by centrifugation and then transferred to XAS sample holders, followed by freezing in liquid nitrogen and later collection of X-ray spectra at 20 K (see SI). The metal content was determined by X-ray fluorescence analytics (Table 1, Supplementary Table 1), revealing 65 ± 19 Mn ions per initially Mn-depleted PSII after illumination in the presence of 240 μM

MnCl$_2$. The calcium content in the PSII-formed Mn oxide could not be reliably determined because CaCl$_2$ was present in the buffer and Ca is known to bind nonspecifically to the used PSII membrane particle preparation (Supplementary Table 1)[43].

**X-ray spectroscopy reveals extended Mn(III/IV) oxides**. The shape of the XANES (X-ray absorption near-edge structure) pronouncedly differed from hexaquo-Mn$^{2+}$, micro-crystalline Mn oxides (Mn$^{III}_2$O$_3$, Mn$^{II,III}_3$O$_4$, β-Mn$^{IV}$O$_2$), and native PSII, but was similar to layered Mn(III,IV) oxides denoted as birnessite[44–46] (Fig. 5a). The K-edge energy indicated a mean redox level of about +3.5, suggesting equal amounts of Mn(III) and Mn (IV) ions (Fig. 5a, Supplementary Fig. 12). EXAFS (extended X-ray absorption fine structure) analysis revealed the atomic structure of the PSII-bound Mn oxide (Fig. 5b, Supplementary Fig. 14; Supplementary Table 2). The EXAFS of Mn-depleted PSII with bound Mn oxide closely resembled birnessite in showing a similar main Mn-O bond length (~1.90 Å), minor longer Mn-O bond length contributions (~2.30 Å, assignable to Jahn-Teller elongated Mn-O distances of Mn$^{III}$ ions) as well as similar main and minor Mn-Mn distances (~2.88 Å, ~3.45 Å). Also, longer Mn-Mn distances (~5.00 Å, ~5.54 Å) were similar. On the other hand, the EXAFS spectra differ clearly from Mn(III)$_2$O$_3$ and β-Mn(IV)O$_2$. The metrical parameters from EXAFS simulations are in good agreement with earlier data for the here studied and related Mn-oxide species of the birnessite-type[21,22,25,47,48]. We note that the long-range order in the oxide particles produced by PSII even exceeds that of the herein used reference oxides of the birnessite-type, as indicated by the magnitudes of the Fourier peaks assignable to the 2.87 and 5.54 Å distances, verifying formation of a comparably extended and well-ordered Mn oxide. Notably, according to the similar XAS spectra, a similar birnessite-type Mn(III,IV)-oxide was formed (i) in the presence as well as absence of CaCl$_2$ in the illumination buffer and (ii) at pH-values of 8.5 as well as 7.0 (Fig. 5 and Supplementary Fig. 15). The number of 50-100 Mn ions per PSII suggests that the Mn oxides could be bound to PSII in form of small nanoparticles (<2 nm).

**Mn-oxide nanoparticles are bound to the PSII core complex**. In the experiments reported above, PSII membrane particles were investigated. These are comparably large membrane fragments containing numerous PSII units per fragment and therefore can be collected by centrifugation at comparably low speed (20,000 × g for 10 min). These centrifugation conditions do not allow for pelleting of unbound oxide nanoparticles of only about 100 Mn ions (diameter <2 nm), suggesting that the Mn-oxide nanoparticles are bound to the PSII membrane particles. To exclude that the Mn-oxide nanoparticles were bound to the lipid bilayer membrane or trapped between stacked membrane sheets, we also investigated detergent solubilized, membrane-free PSII particles using the same protocol for light-induced Mn-oxide formation as used for the PSII membrane particles. Speciation by EXAFS spectroscopy (Supplementary Fig. 16) verified that the same Mn oxide is formed also for detergent-solubilized PSII, thereby providing support for association of Mn-oxide nanoparticles directly with the PSII proteins.

## Discussion

**Mn-oxide formation by PSII**. We have obtained the first direct experimental evidence that PSII devoid of Mn$_4$CaO$_5$ is capable of forming Mn(III,IV)-oxide particles of the birnessite type by light-driven oxidation of Mn$^{2+}$ ions. Presumably these are nanoparticles of 50–100 Mn ions that are bound to the PSII protein complex, as suggested by their presence in PSII membrane

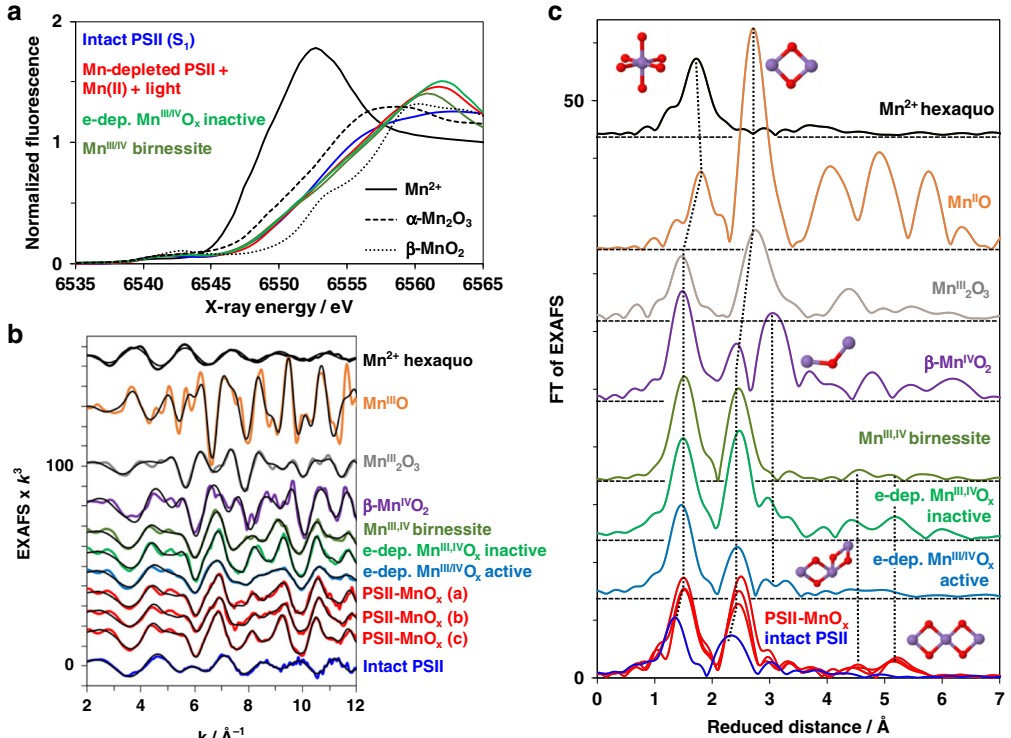

**Fig. 5 X-ray absorption spectroscopy evidencing PSII-bound Mn oxide of the birnessite-type. a** XANES spectra of the native PSII with its active-site Mn $(III)_2Mn(IV)_2CaO_5$ cluster (blue line) and Mn-depleted PSII after the light-driven formation of Mn oxide (red lines) are compared to the spectra of $Mn^{2+}$ in solution (solid black line), electrodeposited Mn oxide (green line), synthetic birnessite (dark green line) and further Mn oxides (dashed and dotted black lines). **b** The corresponding EXAFS spectra in $k$-space as in **a** as well as spectra of further Mn compounds. **c** Fourier-transformed (FT) EXAFS spectra (as in **b**) of: $Mn^{2+}$ ions in solution, indicated reference Mn oxides, the $Mn_4CaO_5$ cluster in intact PSII, and Mn-depleted PSII after illumination in the presence of $Mn^{2+}$. Structural motifs corresponding to the individual peaks are schematically shown; FT peaks relating to the same structural motif are connected by dotted lines. Mn-depleted PSII particles (20 µg mL$^{-1}$ chlorophyll) were illuminated for 3 min (1000 µE m$^{-2}$ s$^{-1}$) in the presence of 240 µM MnCl$_2$ and 60 µM PPBQ (buffer conditions: 1 M glycine-betaine, 15 mM NaCl, 5 mM CaCl$_2$, 5 mM MgCl$_2$, 25 mM MES buffer, pH 8.5; similar data for pH 7.0 is shown in Supplementary Fig. 15) and subsequently collected by centrifugation. (The spectrum shown in **a** as a red line represents the mean of the three spectra from individual samples shown in **b** and **c** as red lines as obtained after subtraction of a 5 % aqueous $Mn^{2+}$ contribution. See Supplementary Table 2 for details and further data.) Source data are provided as a Source Data file.

particle pellets, due to co-sedimentation using a comparably mild centrifugation protocol, as well as in solubilized, membrane-free PSII core particles after their precipitation. Is it possible that larger Mn-oxide particles are formed (e.g., several thousand Mn ions), which would sediment also at moderate centrifugation speed? The cooperation of many PSII centers in formation of a single large Mn-oxide nanoparticle is unlikely, inter alia because efficient electron transfer from $Mn^{2+}$ ions to the redox-active tyrosine cannot occur over distances that are as long as the distance between neighboring PSII dimers in PSII membrane particles and even more so for solubilized PSII. Similarly, also the fast spontaneous fusion of Mn-oxide nanoparticles to more extended oxide particles is highly unlikely. Aggregation mediated by non-bonding interactions cannot be rigorously excluded, but is disfavored by the expected concentrations of oxide particles in the sub-micromolar range. On these grounds and supported by the inhibitory effect of Mn-oxide formation on electron donation (Fig. 4 and Supplementary Fig. 8), we assume that Mn-oxide nanoparticles are bound to the PSII core complex, likely in the vicinity of the redox-active tyrosine (Y$_Z$ in Fig. 1).

Light-driven $Mn^{2+}$ oxidation also can promote self-assembly of the functional $Mn_4CaO_5$ complex, which is a comparably inefficient (low quantum yield) low-light process denoted as photoactivation[19,20]. Cheniae et al. investigated photoactivation and observed light-driven binding of about 18 membrane-bound Mn ions per PSII if Ca ions were excluded from the

photoassembly buffer[49], whereas we here observe binding of clearly more Mn ions (65 ± 19 Mn ions, Table 1), irrespective of the absence or presence of Ca ions at a moderate concentration (5 mM) in the photoassembly buffer (Supplementary Fig. 14). The presence of 5 mM CaCl$_2$ allows for photoactivation, although a higher concentration is required for optimal photoactivation yield[49,50]. The about 20 times higher light intensities we used likely promoted the oxidation and binding of numerous Mn ions at the expense of formation of a single native $Mn_4CaO_5$ cluster, because the latter requires low-light intensities presumably due to the presence of a slow 'dark rearrangement' step for assembly[19].

Since Cheniae's work[49], it had remained an open question in what form a larger number of Mn ions can bind to PSII membrane particles. Coordination of individual high-valent Mn ions to protein groups is one possibility (as often observed for divalent cations and trivalent Fe ions); the formation of extended protein-bound Mn-oxide nanoparticles is another possibility. Under our high-light conditions, Mn(III,IV)-oxide formation clearly is dominant. The Mn-oxide cluster size seems to be limited to around 100 Mn ions, which may correspond to a nanoparticle of about 20 Å in diameter (Fig. 6a). Such a particle may well be formed within the PSII cavity that becomes solvent-exposed upon removal of the extrinsic protein subunits (Fig. 1). These subunits are evolutionarily younger than the PSII core proteins[51] and are absent in related anoxygenic photosystems[52]. Thus, it is well conceivable that an early PSII ancestor would have lacked these

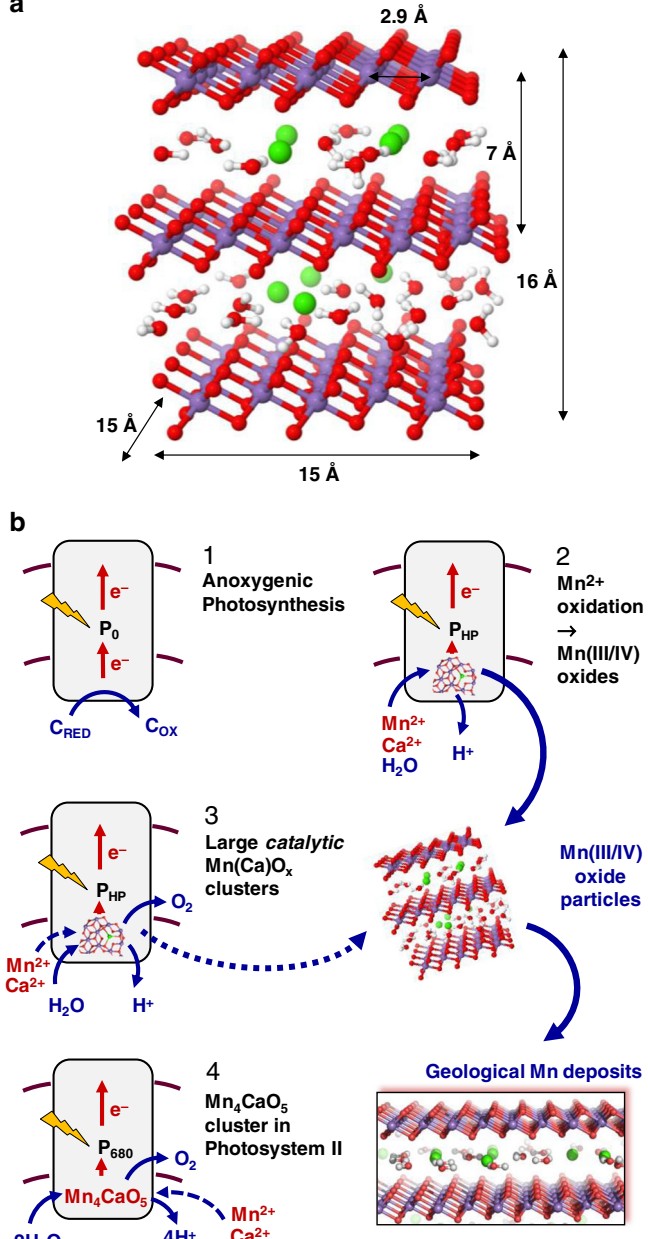

**Fig. 6 Evolution of oxygenic photosynthesis and relation to geological Mn-oxide deposits. a** Structural model of a birnessite fragment with 64 Mn ions (based on published atomic coordinates, protons were added for illustration only). Atom color coding: violet, Mn(III/IV) ions; red, O or OH; green, $Ca^{2+}$; gray, H. **b** Proposed sequence of evolutionary events. (1) Starting from an early anoxygenic photosystem in a cyanobacterial ancestor, either with a low-potential or a high-potential primary donor, (2) a variant with suitable donor potential ($P_{HP}$) and metal-binding site facilitated $Mn^{2+}$ oxidation to generate protein-bound Mn(III,IV)-oxide clusters, which upon sedimentation contributed to geological Mn-oxide deposits. (3) The PSII-bound clusters developed into primordial water-oxidizing and $O_2$-evolving catalytic complexes, which, (4) due to a dedicated binding site, were down-sized to the present $Mn_4CaO_5$ catalyst. At some point, Mn-oxide particles also may have been involved in a quasi-respiratory cycle.

extrinsic proteins and therefore could accommodate a Mn-oxide nanoparticle. Furthermore, an ancient autotroph, capable of exploiting $Mn^{2+}$ as a metabolic reductant[32,33], would be expected to be configured so that the donor side of the early PSII ancestor

would be exposed to the environment as opposed to being sequestered within the lumen of modern thylakoids. In this context, the cyanobacterium *Gloeobacter violaceus* provides an interesting example. *Gloeobacter* occupies a basal phylogenetic position and evolved before the appearance of thylakoids. It possesses photosynthetic reaction centers that are located in the cytoplasmic membrane with the oxidizing domain of PSII facing the periplasmic space and thus the exterior of the cell[53]. Thus *Gloeobacter* provides an example of how a primordial reaction center might have been arranged to facilitate the photochemical utilization of $Mn^{2+}$ as a reductant source, as originally proposed by Zubay[32].

**Relation to water-oxidizing synthetic Mn oxides.** Birnessite and buserite are layered, typically non-crystalline metal-oxides with sheets of edge-sharing $MnO_6$ octahedra (which corresponds to di-μ-oxo bridging between neighboring Mn ions) separated by water and cations, e.g., $Na^+$ or $Ca^{2+}$, in the interlayer space[44,45,54]. Birnessite and buserite differ regarding the number of water-cation layers in between two oxide layers (one in birnessite, two in buserite), but share the same fundamental structure of the Mn(III, IV) oxide layers and thus are often jointly denoted as birnessite-type Mn oxides. Notably, a Mn oxide denoted as ranciéite is isostructural to birnessite and contains Mn and Ca ions at approximately the same 4:1 stoichiometry as present in the $Mn_4CaO_5$ cluster of PSII[55], suggesting a possible relation[27,56–58]. Birnessite-type Mn oxides are a major component of Mn-oxide ocean nodules[44] and biogenic Mn oxides[59]. Their diagenetic reductive conversion to Mn-bearing carbonates, on geological time scales, may explain the early Mn deposits reported by Johnson et al.[33]. Notably, by electrodeposition and other synthesis protocols, Mn(III/IV) oxides of the birnessite type can be formed that are either active or largely inactive in water oxidation, depending on their atomic structure[21,22,25,47,60,61]. These Mn(III/IV) oxides share key features with the $Mn_4CaO_5$ cluster of the biological catalyst, including joint structural motifs and facile oxidation state changes during catalytic operation[24]. The presence of Ca ions is especially favorable for water oxidation activity by synthetic manganese oxides, pointing towards similar water-oxidation mechanisms in the synthetic oxides and the biological $Mn_4CaO_5$ cluster of PSII[21,22,25,62]. Regarding their high degree of structural order, the Mn-oxide particles formed by PSII resemble electrodeposited Mn oxides that are able to undergo Mn(III)—Mn(IV) redox transitions, but exhibit low electrochemical water oxidation activity[24]. The structural characteristics that have been identified for transforming a largely inactive Mn oxide into an oxide with sizeable water-oxidation activity[24] are apparently lacking in the Mn-oxide particles formed by PSII, which may explain the absence of detectable water-oxidation activity by the herein investigated PSII-bound Mn-oxide particle.

**Mechanism of Mn-oxide formation.** The basic biochemical mechanism of the here described light-induced Mn-oxide formation likely involves the initial binding of $Mn^{2+}$ ions followed by Mn oxidation and stabilization of the oxidized Mn(III/IV) ions by di-μ-oxo bridging, in analogy to both the photoassembly process of today's $Mn_4CaO_5$ cluster[19,20] and the oxidative self-assembly process in the electrodeposition of non-biological birnessite-type Mn oxides[25,47]. The formation of extended oxide particles likely involves a nucleation-and-growth mechanism. In the photosystem, the initial site of $Mn^{2+}$ binding and formation of an oxide nucleus likely is provided by carboxylate and possibly imidazole sidechains of protein residues followed by an oxide growth that does not require further ligating residues.

**Oxide-incorporation hypothesis on the evolution of today's $Mn_4CaO_5$ cluster**. Various hypotheses on the origin of the $Mn_4CaO_5$ cluster have been proposed. According to Raymond and Blankenship the interaction of an anoxygenic PSII with a manganese catalase, utilizing hydrogen peroxide as the source of electrons, led to the formation of today's $Mn_4CaO_5$ cluster[28], without invoking Mn oxides. Dismukes and coworkers developed hypotheses on the evolution of oxygenic photosynthesis by focusing on the inorganic chemistry of Mn and bicarbonate[63]; their analyses could complement the herein developed ideas on the evolutionary role of Mn oxides in the future. In 2001, Russell and Hall developed their influential Mn-oxide incorporation hypothesis[56]. They suggested that a 'ready-made' cluster must have been co-opted whole by a (mutant?) protein[57,58]. Russell and Hall specifically proposed that dissolved $Mn^{2+}$ ions were photo-oxidized at extremely short wavelengths[64] to colloidal clusters of $[CaMn_4O_9 \cdot 3H_2O]$, which are closely related to the birnessite-type Mn oxide denoted as rancicíéite. Incorporation of this or a similar ready-made $Mn_4Ca$ species into a PSII ancestor would have led to the $Mn_4CaO_5$ cluster of today's PSII. This hypothesis is in line with analyses of Yachandra and Sauer who systematically compared the structural relations between various Mn oxides and the biological metal complex, revealing intriguing similarities[27] (for further discussion, see SI Appendix— Supplementary Note).

**Alternative hypothesis on the evolution of the $Mn_4CaO_5$ cluster**. We see a close relation between inorganic Mn oxides and today's $Mn_4CaO_5$ cluster of PSII that differs distinctively from the Mn-oxide incorporation hypotheses outlined above. In our study, the facile formation of birnessite-type Mn- oxide particles by PSII is reported. They (i) share structural motifs with the biological cluster in PSII[10,65] and biogenic Mn oxides in general[59,66] and (ii) resemble synthetic Mn oxides closely that have been investigated as synthetic catalyst materials[24]. On these grounds we propose a scenario illustrated in Fig. 6: Rather than Mn-oxide incorporation, Mn-oxide nanoparticles were formed by an evolutionary precursor of PSII, inter alia enabling the formation of early geological Mn-oxide deposits. Initially, dissolved $Mn^{2+}$ ions served as a source of reducing equivalents eventually needed for $CO_2$ reduction, as has been suggested first by Zubay[32] and later by others[31,33]. At a later stage, down-sized oxide particles developed into today's water-oxidizing $Mn_4CaO_5$ cluster. Whether the early photosynthetic reaction centers initially exhibited a sufficiently high potential for the oxidation of aqueous $Mn^{2+}$ ions[67,68] or such a potential was acquired during evolution starting from a low-potential ancestor[31] has to remain an open question at the present stage (Fig. 6). Olson has already favored the high-potential first hypothesis 50 years ago[68], and recently such ideas have gained support based on structural and genomic comparisons from Cardona and coworkers[69]. We note that our hypothesis that Mn-oxide incorporation precedes the formation of the present water oxidation catalyst in PSII is independent on the earlier way of evolution of a high-potential reaction center, because such a species is needed for both processes.

Are there evolutionary relics that may support our above hypothesis? Extensive studies on the diversity of the PSII reaction center protein D1 have revealed several atypical variants that can be distinguished phylogenetically[70,71]. These early evolved forms lack many residues needed for the binding of today's $Mn_4CaO_5$ cluster and could relate to ancient Mn-oxide-forming photosystems, even though today they might play other physiological roles (e.g., in the synthesis of chlorophyll $f$[72]).

**Summary of potential evolutionary implications**. The ability for light-driven Mn-oxide formation by an ancient photosystem

represents an important touchstone for evaluation of three interrelated hypotheses that each addresses a remarkable facet of the evolution of the Earth's biosphere and geosphere:

(i) The ability for the direct and facile photosynthetic formation of stable Mn(III/IV)-oxide particles supports that early Mn deposits[33] resulted directly from photosynthetic activity.

(ii) Structural and functional similarities between water-oxidizing synthetic Mn oxides and the here described Mn-oxide formation by PSII suggests that in the evolution of PSII, there may have been a transition from extended Mn-oxide nanoparticles towards the $Mn_4CaO_5$ cluster of today's PSII, as illustrated by Fig. 6.

(iii) An early quasi-respiratory cycle has been proposed that involves the formation of Mn(III/IV) oxide particles followed by utilization of the oxidizing equivalents stored in the Mn oxide for an efficient quasi-respiratory activity in the Archean or early Paleoproterozoic, when the Earth's atmosphere had been essentially $O_2$-free, as detailed in ref. [65].

By showing that today's PSII can form birnessite-type Mn-oxide particles efficiently, even without any specific protein subunits that would support Mn-oxide formation, the general biochemical feasibility is verified. This finding renders it highly likely that similarly also an ancient photosystem, the PSII ancestor, had the ability for the light-driven formation of Mn oxides from hexaquo $Mn^{2+}$ ions. In conclusion, we believe that our successful demonstration of the photosynthetic formation of Mn(III/IV)-oxide particles provides relevant support for the above three hypotheses.

## Methods

**Preparation of PSII membrane particles**. Native PSII-enriched thylakoid membrane particles were prepared from fresh market spinach following our established procedures[35]. Their typical $O_2$-evolution activity (as determined by polarography with a Clark-type electrode at 27 °C) was ~1200 µmol $O_2$ mg$^{-1}$ chlorophyll h$^{-1}$, which proved the full integrity of the PSII proteins and the water-oxidizing $Mn_4CaO_5$ complex. We have shown earlier that this type of PSII preparation contains ~200 chlorophyll molecules per PSII reaction center[73,74]. When kept for prolonged time periods in the dark, the $Mn_4CaO_5$ complex is synchronized in the $S_1$ state of its catalytic cycle, which is established to represent a $Mn(III)_2Mn(IV)_2$ oxidation state[75,76].

**Mn-depletion of PSII**. Removal of $Mn_4CaO_5$ and of the three extrinsic proteins of PSII (PsbQ, PsbP, and PsbO) was carried out using a literature procedure and evaluated using metal quantification and gel electrophoresis (see below)[36]. PSII membranes were dissolved at 200 µg chlorophyll mL$^{-1}$ in a high-salt buffer (30 mL) containing 20 mM TEMED (N,N,N',N'-tetramethylethylenediamine) as a reductant for the PSII-bound Mn(III/IV) ions, 20 mM MES (2-(N-morpholino)ethane-sulfonic-acid) buffer (pH 6.5), and a high-salt concentration (500 mM $MgCl_2$) and incubated in the dark on ice for 10 min. PSII membranes were pelleted by centrifugation (Sorvall RC26, 12 min, 50,000 × $g$, 4 °C), the pellet was three times washed by dissolution in a buffer (30 mL) containing 35 mM NaCl and 20 mM TRIS (tris (hydroxymethyl)aminomethane) buffer (pH 9.0) and pelleting by centrifugation as above, and the final pellet of Mn-depleted PSII membranes was dissolved at ~1 mg chlorophyll mL$^{-1}$ in a buffer containing 1 M glycine-betaine, 15 mM NaCl, 5 mM $CaCl_2$, 5 mM $MgCl_2$, and 25 mM MES buffer (pH 6.3). The PSII preparations (~2 mg chlorophyll mL$^{-1}$) were thoroughly homogenized by gentle brushing and frozen in liquid nitrogen for the spectroscopic experiments. The Mn-depleted PSII showed zero $O_2$-evolution activity as revealed by polarography.

**TXRF analysis**. X-ray emission spectra were recorded on a Picofox instrument (Bruker) and metal contents of PSII samples were determined from the data using the (fit) routines available with the spectrometer[39]. PSII membranes were adjusted to a chlorophyll concentration of 1–2 mM and to a 20 µL aliquot, a gallium concentration standard (1 mg mL$^{-1}$, 20 µL; Sigma-Aldrich) was added, and samples were homogenized by brief sonication (see Supplementary Fig. 1). A 5 µL aliquot of the samples was pipetted on clean quartz discs for TXRF, dried on a heating plate, loaded into the spectrometer, and TXRF spectra were recorded within 10–30 min. At least three repetitions of each sample and three independently prepared samples of each PSII preparation were analyzed. The TXRF data on Mn-oxide formation by Mn-depleted

PSII shown in Table 1 were obtained using the same illumination and centrifugation protocol also used for preparation of the X-ray spectroscopy samples of Fig. 5.

**Optical absorption spectroscopy and illumination procedures.** For the optical absorption spectroscopy experiments, stock suspensions of the PSII preparations were diluted at 20 µg chlorophyll mL$^{-1}$ (~0.1 µM PSII centers) in a buffer (3 mL) containing 1 M glycine-betaine, 15 mM NaCl, 5 mM CaCl$_2$, 5 mM MgCl$_2$, and 25 mM MES buffer (pH 6.3–8.5) and reactants (oxidized 2,6-dichlorophenol-indophenol = DCPIP$^{ox}$ from Fluka, other electron acceptors as in Supplementary Fig. 11, MnCl$_2$) were added at indicated concentrations. The pH was routinely controlled prior to and after the illumination assays in the actual cuvette and found to be stable within ±0.2 pH units within a time period of at least 20 min. Optical absorption spectra of the samples in a 300–900 nm range were recorded within about 10 s at given time intervals (about 0.3–1.0 min) in a 3 mL quartz cuvette (Helma QS1000, 1 cm pathlength) using a Cary 60 spectrometer (Agilent). Alternatively, time traces of absorption were recorded at selected wavelengths (i.e., 604 nm to monitor DCPIP$^{ox}$ reduction) for up to 30 min. Temperature logging revealed that the sample temperature varied by <2 °C within the extended illumination periods. PSII-sample filled cuvettes in the spectrometer were continuously illuminated from the top side using a white-light lamp (Schott KL1500, halogen light bulb with cold-light reflector) with attenuation option, which was directed through a heat-protection filter (Schott KG5) to the cuvette by a ~20 cm light-guide (the full cuvette volume was homogenously illuminated). The combination of light source, KG5 filter, light guide, and cuvette material resulted in an effective spectral range of about 400–750 nm (limits correspond to the 10% level) effectively excluding UV irradiation, thereby minimizing potential interferences due to peroxide formation resulting from direct Mn-oxide excitation, and sample heating due to infrared light (thermal radiation). Several spectra (or time points) were recorded in the dark (prior to and after addition of, e.g., DCPIP), the light was switched on (or off) at indicated time points, and data were recorded on a PC linked to the spectrometer. Evaluation and fit analysis of absorption data was carried out using the Origin software (OriginLab). Light intensities at the sample center position were determined using a calibrated sensor device inserted in the spectrometer.

**X-ray absorption spectroscopy.** XAS at the Mn K-edge was performed at beamline KMC-3 at the BESSY-II synchrotron (Helmholtz Zentrum Berlin) with the storage ring operated in top-up mode (250 or 300 mA), using a standard set-up as described in refs. [25,77]. A double-crystal Si[111] monochromator was used for energy scanning, the sample X-ray fluorescence was monitored with an energy-resolving 13-element germanium detector (Canberra) or a 13-element silicon-drift detector (RaySpec), and samples were held in a liquid-helium cryostat (Oxford) at 20 K (in a 0.2 bar He heat-exchange gas atmosphere at an angle of 55° to the incident X-ray beam). The X-ray spot size on the sample was shaped by slits to about 1 (vertical) × 5 (horizontal) mm$^2$, the X-ray flux was ~10$^{10}$ photons s$^{-1}$, the EXAFS scan duration was ~10–20 min. The energy axis was calibrated (±0.1 eV accuracy) using a Gaussian fit to the pre-edge peak (6543.3 eV) in the transmission spectrum of a permanganate (KMnO$_4$) powder sample, which was measured in parallel to the PSII samples. For XAS data evaluation, up to 30 deadtime-corrected, energy-calibrated ($I/I_0$) XAS monochromator scans (each on a fresh sample spot) were averaged and normalized XANES and EXAFS spectra were extracted after background subtraction using in-house software[78]. EXAFS simulations in $k$-space were carried out using in-house software (SimX) and scattering phase functions calculated with FEFF9.0[79] ($S_0^2 = 0.8$). Calculation of the filtered $R$-factor ($R_F$, the difference in % between fit curve and Fourier-backtransform of the experimental data in a 1–5 Å region of reduced distance)[80] facilitated evaluation of the EXAFS fit quality. Fourier-transforms of EXAFS spectra were calculated with cos windows extending over 10% of both $k$-range ends.

**Sample preparation for XAS.** Powder samples of manganese reference compounds (Mn oxides) were prepared from commercially available chemicals (MnCl$_2$, Mn oxides) or from material (birnessite) that was kindly provided by the group of P. Kurz (Uni. Freiburg, Germany), diluted by grinding with boron-nitride (BN) to a level, which resulted in <15% absorption at the K-edge maximum to avoid flattening effects in fluorescence-detected XAS spectra, loaded into Kapton-covered acrylic-glass holders, and frozen in liquid nitrogen. Aqueous MnCl$_2$ (20 mM) samples were prepared at pH 7.0. Unless otherwise specified, PSII samples were prepared as follows: Mn-depleted PSII samples (3 mL) were prepared similar to the samples for optical absorption spectroscopy (see above), the pH was adjusted to the desired value (pH 7 or 8.5), and samples were illuminated for 3 min at 1000 µE m$^{-2}$ s$^{-1}$ or kept in the dark as a control after addition of 240 µM MnCl$_2$ and 60 µM PPBQ$^{ox}$. Thereafter, the cuvette volume was rapidly mixed with ice-cold MES buffer (7 mL, pH 7 or 8.5, see above for ingredients) on ice in the dark, the pH was measured using a pH electrode and, if necessary, readjusted to the desired value (+/-0.1 pH units), the PSII membranes were pelleted by centrifugation (10 min, 20,000 × g, 2 °C), and kept on ice. Several of these sample types were rapidly merged on ice in the dark by loading (~30 µL) into XAS holders, which were immediately frozen in liquid nitrogen. Native PSII samples were prepared by pelleting of dark-adapted O$_2$-evolving PSII membrane particles (~8 mg chlorophyll mL$^{-1}$, pH 6.3), loading of the

pellet material into XAS holders, and freezing in liquid nitrogen[76]. The shown XAS data for the electrodeposited Mn oxides has been collected in the context of earlier studies[21,22,47] and replotted.

**Solubilization of PSII membrane particles to yield membrane-free PSII complexes.** Mn-depleted PSII membrane particles were solubilized as described by Haniewicz et al.[81]. In brief, 20 mM β-dodecylmaltoside (β-DM) was added to the membranes with a chlorophyll concentration of 2 mg mL$^{-1}$. After incubation for 30 min at 4 °C in the dark, the solubilized material was separated from the insoluble fraction by centrifugation at 40,000 × g for 20 min at 4 °C. From the supernatant containing the solubilized Mn-depleted PSII, XAS samples were prepared as follows. An aliquot of the solution with solubilized PSII was added to MES buffer (pH 7.0) containing 240 µM MnCl$_2$, 5 mM CaCl$_2$, 60 µM PPBQ, and 0.03 % (v/v) β-DM to a final chlorophyll concentration of 20 µg mL$^{-1}$ and the sample was illuminated for 1 min at 1000 µE m$^{-2}$ s$^{-1}$. To remove unbound Mn and electron acceptor, the sample was precipitated by adding polyethylene-glycol (PEG 6000) to a final concentration of 4.4% (w/v) and a subsequent centrifugation step (16,000 × g, 10 min, 4 °C)[82]. The pellet was resuspended in MES buffer (pH 7.0) and the PEG precipitation and centrifugation procedure was repeated twice. After the final washing step, the pellet was resuspended in a small volume of buffer and transferred to the sample holder as described above.

**Reporting summary.** Further information on research design is available in the Nature Research Reporting Summary linked to this article.

## Data availability

All data needed to support the conclusions of this manuscript are included in the main text and SI Appendix. The source data underlying Figs. 3–5 are provided as a Source Data file. Source data are provided with this paper.

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

## Acknowledgements

We thank I. Zizak, G. Schuck, and colleagues (Helmholtz Zentrum Berlin) for technical support in the X-ray experiments at the BESSY synchrotron. We thank P. Kurz (Universität Freiburg) for providing samples of birnessite and buserite. We gratefully acknowledge financial support by the Deutsche Forschungsgemeinschaft (DFG, German Research Foundation) within SFB 1078 (project A4) and in form of an Emmy Noether project awarded to D. Nürnberg. The Einstein Foundation Berlin supported this research by an Einstein Fellow project awarded to R. Burnap. Moreover, this study has been funded by the Deutsche Forschungsgemeinschaft under Germany´s Excellence Strategy – EXC 2008/1 – 390540038.

## Author contributions

H.D. designed the study. P.C., S.F., J.H., N.O., R.A., B.Y., and D.J.N. performed the experiments. P.C., M.H., and I.Z. analyzed the data. R.L.B., D.J.N., M.H., and H.D. wrote the manuscript.

## Funding

## Competing interests

The authors declare no competing interests.
