## [Peer Review File · Nature Communications]

Reviewers' comments:

Reviewer #1 (Remarks to the Author):

The ms by Chernev et al describes the identification of birnessite-type nanocrystals formed by light-mediated oxidation of dissolved Mn on PSII depleted of the extrinsic subunits and of the D1-bound manganese cluster. These studies resolve decades-old questions and are of high pertinence to the problem of the origin and evolution of photosynthetic water splitting. The experimental evidence is exhaustive and convincing, and the manuscript is well-written. Due to the relevance with respect to the dispute on the origin of the bioenergetic process which overturned Earth's geo-environment some 3 billion years ago and which undoubtedly prepared the ground for the emergence of complex life, I consider this ms as well suited for the wider audience targeted by NatComm.

I did somewhat agonise over the peculiar kinetics of DCPIP reduction at low concentrations of MnCl₂ in figure 4 but have to admit that I couldn't come up with a good explanation either. I have the feeling, though, that the processes involved in this kinetics merit further attention (not for this ms but as a suggestion to follow up this riddle in the future).

My only point of criticism is that I see a gap in the string of evolutionary events necessary in the authors' scenario which is to my mind overlooked. The authors observe that (present day) PSII oxidises (under appropriate conditions) MnCl₂ to Mn(III)/Mn(IV) mixed oxides in the form of layered minerals and they conclude that this may also have happened in an ancestral pre-PSII which used Mn(II) as electron donor and which somehow derived from an anoxygenic reaction centre. Modern PSII no doubt is thermodynamically able to perform this reaction since the oxidising power of P680+ largely exceeds +1V. Anoxygenic RCs of the type considered to still resemble the ancestral forms don't and the ones we know linger between +150 and +350 mV for their P/P+ potentials. The oxidation of dissolved Mn to its oxides occurs at nominal potentials of roughly +400 mV. You therefore require some kind of big leap in the oxidising potential of your photooxidised Chlorophyll (supposed to have been Bacteriochlorophylls in the respective scenarios) to be able to produce your Mn(III)/Mn(IV) mixed oxides (as also suggested by the work published by some of the authors on the electrodeposition of mixed Mn-oxides). One might want to see a few phrases lost on this topic. NB: Present day PSII IS very oxidizing. The problem I raised arises from the assumption in your scenario that PSII derives from anoxygenic RCs oxidising sulphur compounds, Fe²⁺ and the likes. However, there are also scenarios out there which stipulate that it all started with a highly oxidising RC and then evolved towards all others. Such scenarios go back almost 50 years (John Olson's for example) and are nowadays floated by Tanai Cardona and colleagues. I admit that I was always very reticent with respect to PSII-first hypotheses but the authors' data somewhat shake my convictions ...

Typos:

-page 3, line 11: developed

-page 3, line 14: insufficient

-SI, page 18, second paragraph: tightly bound

and call me pedantic but "noteworthy" is an adjective. You use it as an adverb throughout the ms. This should read "noteworthyly"

Reviewer #2 (Remarks to the Author):

The present study by Chernev et al. is an intriguing piece of work on the photoassembly of a large Mn oxide nanoparticle in a natural water oxidizing enzyme, photosystem II (PSII). PSII usually forms a Mn₄CaO₅ cluster as a water-oxidizing photocatalyst on its electron donor side by a light-driven process called photoactivation. The molecular mechanism of photoactivation remains largely unknown and has been a topic of extensive studies. It was previously shown that in the absence of Ca²⁺, the procedure of photoactivation produced a nonfunctional Mn complex with a higher number of high-valent Mn ions (ref. 48), but no one has pursued this phenomena of nonfunctional

Mn oxide formation. The authors performed the detailed analysis of the Mn oxide complex produced in PSII using UV-Vis absorption, X-ray absorption, and total reflection X-ray fluorescence measurements, and for the first time showed that this is a birnessite-type Mn(III,IV)-oxide nanoparticle comprising 50-100 Mn ions. The data were carefully analyzed and the obtained conclusion was convincing. It is fascinating to know that Mn-depleted PSII functions as a natural photocatalyst for production of the Mn oxide nanoparticle, which is identical to the one produced by electron deposition in an electrochemical cell. The discussion for the implications of their findings in the evolutionary aspect of biogenesis of the Mn₄CaO₅ cluster and further geological aspect on the formation of ancient Mn deposits is well written and insightful. The reviewer thus recommends the publication of this study in Nature Communications basically as it is but after the following minor points are addressed.

1. Figure 1 (A): The structure of PDB 5XNL by Su et al. is obtained not by X-ray crystallography but by cryo-EM.
2. Figure 1 caption, p. 3, line 2 from bottom; p 12, line 32: 18, 24, and 33 kDa are rather old naming of the extrinsic proteins. It is better to use PsbQ, PsbP, and PsbO because the molecular weights are not necessarily accurate.
3. p. 3, last line: Please define the abbreviation of total reflection X-ray fluorescence (TXRF) here.
4. Figure 5 (A) caption: Mn²⁺ solution (solid black line)... further Mn oxides (dashed and dotted black lines).

Reviewer #3 (Remarks to the Author):

The manuscript by Petko Chernev and coworkers deals with important questions of how early photosynthetic organisms have developed Photosystem II for water oxidation, one of the most fundamental bioprocesses on earth, and how they possibly contributed to build up early geological manganese deposits. To decipher these questions, they have analyzed the oxidation products of Photosystem II membrane particles from plants, depleted of the water splitting complex, in presence of the exogenous electron donor Mn(II) and electron acceptors. They found that such prepared particles produced Mn(III/IV)oxides which biomineralized to nanoparticles in a few minutes under high light conditions. The XANES and EXAFS spectra of this product have close similarity to the spectra of synthetic Mn(III/IV)oxides of the birnessite type of which some are capable to oxidize water, although not the one observed in this work. These observations led the authors to propose the interesting and challenging proposition that such nanoparticles were produced and fixed by early photosystems and thus contributed to build up geological manganese deposits and then were downsized to the today's water splitting complex.

The manuscript is clearly and concise in presenting the main results and propositions but lacks some additional experimental evidence, which should make these hypotheses more reliable. The only experimental evidence presented in the work that the Mn oxide nanoparticles are fixed by Photosystem II is the observation that the reduction of the electron acceptor DCPIP stops after a few minutes of high light exposure in presence of high concentration of Mn(II). However, low quantum yield of Mn(II) oxidation and high light could stop DCPIP reduction due to photoinhibition of the donor side of Photosystem II. Mn(III) is known to be highly reactive and could also damage the donor side if continuously produced under high light. Absorption spectra are insufficient to detect these damages so that other control experiments should be presented to exclude these possibilities. In addition, Mn oxide particles can produce harmful hydrogen peroxide under UV light. The Schott KG5 filter used in this work for illumination might be insufficient.

The authors find that the Mn-oxide cluster size are limited to around 100 Mn ions, which may correspond to a nanoparticle that could fit within the Photosystem II cavity. The Mn/PSII ratios of table S1 were obtained by an estimation that all the Photosystem II centers are active during the measurement time. If inhibitory effects are present, the calculated cluster size would be much bigger and could not fit anymore into the Photosystem II cavity possibly compromising the authors statement that these clusters are biomineralized and fixed by Photosystem II.

Minor comments to the manuscript are:

-The use of buffer and pH in this work is quite confusing. All experiments use Mes as buffer with pH adjusted to 7, 7.5 and 8 although Mes has no buffering capacities from pH 7.5. While all pH values are used to present optical measurements, XAS measurements (Figure 5, S10, S11) use only Photosystem II particles prepared at pH 8. No comments in the manuscript are made on the choice and influence of pH and buffer to the measurements and, particularly, to the production of Mn oxide nanoparticles.

-In Figure S9, the efficiency of light induced Mn oxide production is compared for different exogenous electron acceptors. The slow hydrophilic, but most electro-positive, electron acceptor ferricyanide only produces low amounts of Mn oxides which is in favor of the production of biomineralized nanoparticles at the donor side that necessitates high electron turnover. However, to ensure that the reduced hydrophobic electron acceptors are not implicated in Mn oxide nanoparticle production, a control using a mixture of a hydrophobic electron acceptor and ferricyanide is missing.

-The spectra of Figure S4 don't have much of interest. If at all, it would be more instructive to see the influence of 240 μM MnCl_2

-The experiment of Figure S5 is a control. However, the concentration of DCPIP (45 μM) is not the same as used for other experiments.

In summary, the actual state of the manuscript still lacks clear experimental evidence that the Mn oxide nanoparticles are biomineralized and fixed by Photosystem II which is the essential conclusion of this manuscript and which distinguishes it from other propositions found in literature. If such additional experimental evidence is provided by the authors, it would be worth publishing in Nature communication.

Reply to Reviewers

First of all, we thank all three reviewers for careful consideration of our study and their insightful as well supportive comments. We apologize for the long revision period of four months. Before resubmission, we wanted to complete further supporting experiments specifically in response to some of the valuable suggestions by the reviewers, but could not do so earlier, because of a 3-month Corona shut-down of our laboratories.

Reviewer #1

The ms by Chernev et al describes the identification of birnessite-type nanocrystals formed by lightmediated oxidation of dissolved Mn on PSII depleted of the extrinsic subunits and of the D1-bound manganese cluster. These studies resolve decades-old questions and are of high pertinence to the problem of the origin and evolution of photosynthetic water splitting. The experimental evidence is exhaustive and convincing, and the manuscript is well-written. Due to the relevance with respect to the dispute on the origin of the bioenergetic process which overturned Earth's geo-environment some 3 billion years ago and which undoubtedly prepared the ground for the emergence of complex life, I consider this ms as well suited for the wider audience targeted by NatComm.

◇ We thank the Reviewer for the precise and positive evaluation of our work and recommendation for publication.

I did somewhat agonise over the peculiar kinetics of DCPIP reduction at low concentrations of MnCl₂ in figure 4 but have to admit that I couldn't come up with a good explanation either. I have the feeling, though, that the processes involved in this kinetics merit further attention (not for this ms but as a suggestion to follow up this riddle in the future).

<1b> Indeed, the kinetics of electron transfer to DCPIP (and other electron acceptors) at low Mn concentrations versus the high-Mn²⁺ kinetics that result in formation of Mn oxide nanoparticles represent an interesting riddle that clearly merits attention. Additional data analysis and discussion is now provided in Figs. S8 and S9. Future work on this phenomenon (or rather syndrome of phenomena) is in planning, aiming at completion in 2021 or 2022.

My only point of criticism is that I see a gap in the string of evolutionary events necessary in the authors' scenario which is to my mind overlooked. The authors observe that (present day) PSII oxidises (under appropriate conditions) MnCl₂ to Mn(III)/Mn(IV) mixed oxides in the form of layered minerals and they conclude that this may also have happened in an ancestral pre-PSII which used Mn(II) as electron donor and which somehow derived from an anoxygenic reaction centre. Modern PSII no doubt is thermodynamically able to perform this reaction since the oxidising power of P680+ largely exceeds +1V. Anoxygenic RCs of the type considered to still resemble the ancestral forms don't and the ones we know linger between +150 and +350 mV for their P/P+ potentials. The oxidation of dissolved Mn to its oxides occurs at nominal potentials of roughly +400 mV. You therefore require some kind of

big leap in the oxidising potential of your photooxidised Chlorophyll (supposed to have been Bacteriochlorophylls in the respective scenarios) to be able to produce your Mn(III)/Mn(IV) mixed oxides (as also suggested by the work published by some of the authors on the electrodeposition of mixed Mn-oxides). One might want to see a few phrases lost on this topic. NB: Present day PSII IS very oxidizing. The problem I raised arises from the assumption in your scenario that PSII derives from anoxygenic RCs oxidising sulphur compounds, Fe²⁺ and the likes. However, there are also scenarios out there which stipulate that it all started with a highly oxidising RC and then evolved towards all others. Such scenarios go back almost 50 years (John Olson's for example) and are nowadays floated by Tanai Cardona and colleagues. I admit that I was always very reticent with respect to PSII-first hypotheses but the authors' data somewhat shake my convictions ...

<1b> We agree with the reviewer that this point requires more attention. Therefore, statements on the early-high potential RC vs early-low potential RC discussion, together with references on Olson's and Cardona's work, are now provided in the Discussion section (bottom of pg. 12, top of pg.13) and in the caption of Figure 6. We think that at present, both hypotheses have support and our study does not provide a truly decisive argument that could tilt the balance of evidence to one or the other side. (We note that our hypothesis that light-driven Mn oxide formation has played an important role in early evolution of oxygenic photosynthesis does not critically depend on the outcome of the controversy.)

Typos:

-page 3, line 11: developed

-page 3, line 14: insufficient

-SI, page 18, second paragraph: tightly bound

and call me pedantic but “noteworthy” is an adjective. You use it as an adverb throughout the ms. This should read “noteworthily”

<1c> The typos and wording were corrected.

Reviewer #2

The present study by Chernev et al. is an intriguing piece of work on the photoassembly of a large Mn oxide nanoparticle in a natural water oxidizing enzyme, photosystem II (PSII). PSII usually forms a Mn₄CaO₅ cluster as a water-oxidizing photocatalyst on its electron donor side by a light-driven process called photoactivation. The molecular mechanism of photoactivation remains largely unknown and has been a topic of extensive studies. It was previously shown that in the absence of Ca²⁺, the procedure of photoactivation produced a nonfunctional Mn complex with a higher number of high-valent Mn ions (ref. 48), but no one has pursued this phenomena of nonfunctional Mn oxide formation. The authors performed the detailed analysis of the Mn oxide complex produced in PSII using UV-Vis absorption, X-ray

absorption, and total reflection X-ray fluorescence measurements, and for the first time showed that this is a birnessite-type Mn(III,IV)-oxide nanoparticle comprising 50-100 Mn ions. The data were carefully analyzed and the obtained conclusion was convincing. It is fascinating to know that Mn-depleted PSII functions as a natural photocatalyst for production of the Mn oxide nanoparticle, which is identical to the one produced by electron deposition in an electrochemical cell. The discussion for the implications of their findings in the evolutionary aspect of biogenesis of the Mn₄CaO₅ cluster and further geological aspect on the formation of ancient Mn deposits is well written and insightful. The reviewer thus recommends the publication of this study in Nature Communications basically as it is but after the following minor points are addressed.

<> We thank the Reviewer for the insightful evaluation of the study and for the strong recommendation for publication.

1. Figure 1 (A): The structure of PDB 5XNL by Su et al. is obtained not by X-ray crystallography but by cryo-EM.

<2a> The reviewer is right, now we state “Cryo electron microscopy structure of native plant PSII...” in the Fig. 1 caption. Thank you for this remark.

2. Figure 1 caption, p. 3, line 2 from bottom; p 12, line 32: 18, 24, and 33 kDa are rather old naming of the extrinsic proteins. It is better to use PsbQ, PsbP, and PsbO because the molecular weights are not necessarily accurate.

<2b> We changed the annotation of the extrinsic subunits as requested throughout the text and now state in the Fig. 1 caption: “Omitting the extrinsic subunits (PsbQ, PsbP, and PsbO with approximate molecular weights of 18, 24, and 33 kDa) exposes a cavity (yellow shading, the subunits are absent in our Mn-depleted PSII).”

3. p. 3, last line: Please define the abbreviation of total reflection X-ray fluorescence (TXRF) here.

<2c> TXRF is the abbreviations for "total-reflection X-ray fluorescence", as now said at this text position, and we included a reference for the reader to follow for more information on this useful technique. Briefly, this method facilitates elemental analysis by energy-resolved detection of the characteristic X-ray fluorescence (XRF) of the various elements with improved sensitivity due to use of a total reflection geometry for the exciting X-rays.

4. Figure 5 (A) caption: Mn²⁺ solution (solid black line)... further Mn oxides (dashed and dotted black lines).

<2d> The Fig. 5 caption was edited as requested.

Reviewer #3

The manuscript by Petko Chernev and coworkers deals with important questions of how early photosynthetic organisms have developed Photosystem II for water oxidation, one of the most fundamental bioprocesses on earth, and how they possibly contributed to build up early geological manganese deposits. To decipher these questions, they have analyzed the oxidation products of Photosystem II membrane particles from plants, depleted of the water splitting complex, in presence of the exogenous electron donor Mn(II) and electron acceptors. They found that such prepared particles produced Mn(III/IV)oxides which biomineralized to nanoparticles in a few minutes under high light conditions. The XANES and EXAFS spectra of this product have close similarity to the spectra of synthetic Mn(III/IV)oxides of the birnessite type of which some are capable to oxidize water, although not the one observed in this work. These observations led the authors to propose the interesting and challenging proposition that such nanoparticles were produced and fixed by early photosystems and thus contributed to build up geological manganese deposits and then were downsized to the today's water splitting complex. The manuscript is clearly and concise in presenting the main results and propositions but lacks some additional experimental evidence, which should make these hypotheses more reliable.

◁ We thank the Reviewer for bringing our results so clearly to the point and for the positive evaluation. Overall, we have clarified the key points in the writing and included new experiments to address concerns and provide further evidence for our conclusions as described in the following.

The only experimental evidence presented in the work that the Mn oxide nanoparticles are fixed by Photosystem II is the observation that the reduction of the electron acceptor DCPIP stops after a few minutes of high light exposure in presence of high concentration of Mn(II). However, low quantum yield of Mn(II) oxidation and high light could stop DCPIP reduction due to photoinhibition of the donor side of Photosystem II. Mn(III) is known to be highly reactive and could also damage the donor side if continuously produced under high light. Absorption spectra are insufficient to detect these damages so that other control experiments should be presented to exclude these possibilities.

<3a> We thank the reviewer on this point and we now clarify the argument. Our central evidence for Mn oxide binding to PSII does not come from our observation that DCPIP reduction stops after a few minutes, but from the sedimentation behavior (see below paragraph). We apologize for not having emphasized this point in the originally submitted manuscript. In the revised manuscript the point is (i) clearly stated (first paragraph of Discussion section, pg. 9) and (ii) supported by additional experiments support (last paragraph of Results section, pg. 8), and also detailed in the following.

Centrifugation/sedimentation behavior suggest Mn oxides bound to PSII: The samples for XAS were derived by centrifugation after illumination Mn-depleted PSII in the presence of MnCl₂, thereby pelleting PSII membrane particles. This centrifugation step was done at comparably low speed, that is at 20000 g for 10 min. Sedimentation of the PSII particles occurred because these are comparably large membrane fragments containing numerous PSII

per fragment. The Mn oxide nanoparticles were found in the sedimented fraction of PSII membrane particles, but it can be excluded that sedimentation of much smaller, free (unbound) Mn oxide nanoparticles occurs at these comparably low centrifugation speed and time. (Harvesting small nanoparticles by centrifugation is possible, but typically requires employment of a high-speed ultracentrifuge.)

We note that oxide particles of clearly larger dimensions (> 10 nm) could sediment at low centrifugation speed. However, taking into account the distance between PSII dimers (for membrane particles and even more so for solubilized PSII), the cooperation of many PSII in formation of a single large Mn oxide nanoparticle is highly unlikely because efficient electron transfer from Mn ions to the redox-active tyrosine cannot occur over large distances. Similarly, also the fast, spontaneous fusion of Mn oxide nanoparticles to more extended oxide particles is highly unlikely; aggregation mediated by non-bonding interactions cannot be rigorously excluded, but is disfavored by the expected concentrations of oxide particles in the sub-micromolar range. (The above evidence for Mn oxide bound to PSII is now summarized in the first paragraph of the discussion section, on pg. 9.)

Additional experiments on membrane-free solubilized PSII particles: the following was performed to test the hypothesis that the Mn oxide particles might be bound to the lipid bilayer membranes or trapped between stacked membrane sheets. To assess this possibility, we meanwhile have performed an informative additional experiment: The PSII membrane particles were detergent-solubilized for obtaining membrane-free PSII core complexes. Using the same protocol for light-induced Mn oxide formation, the *EXAFS spectra (Figure S16)* verify that the same Mn oxide is formed also for detergent solubilized PSII, thereby providing additional support for association of Mn oxide nanoparticles directly with the PSII core complex, as now described in the last paragraph of the Results section (on pg. 8, with data shown in Figure S16).

Does photoinhibition influence our results? We proposed that at high MnCl_2 concentrations the formation of Mn oxides is terminated because the already formed Mn oxide nanoparticle prevents further Mn^{2+} oxidation by the redox-active tyrosine denoted as Tyr_Z or Y_Z. The reviewer argues, in a very insightful way, that not the inhibitory effect of bound Mn oxide nanoparticles, but other types of photoinhibitory effects may explain termination Mn oxide formation after about 1 min.

Please keep in mind that, as stated above, our conclusion the Mn oxide particles are bound to PSII does not rely on the mechanism that terminates electron flow to DCPIP at high MnCl_2 concentration but rather on the centrifugation/ sedimentation behavior. Nonetheless, the termination of electron flow at high MnCl_2 is truly interesting. We are aware that prolonged high-intensity illumination of PSII generally results in photoinhibitory damage, even more so in Mn-depleted PSII. However, we do not think that photoinhibitory damage explains the here observed termination of electron donation in the presence of high concentrations of MnCl_2 . *Evidence against photoinhibitory damage of Mn-depleted comes from the continuous rate of DCPIP reduction at low MnCl_2 concentrations that extends over several about 10 min, whereas the Mn oxide formation at high MnCl_2 concentration is terminated after about 1 min.*

We propose that the latter results from inhibition of the Tyrz-mediated electron transport once a certain amount of Mn oxide has been formed (see Fig. S8 added to the SI). The former, that is, continuous electron donation to Mn-depleted PSII at comparably low Mn²⁺ concentration, has been investigated in the last 50+ years repeatedly and shown to be related to Mn binding with a pH-dependent binding constant around 1 μM (often denoted as high-affinity Mn-binding site). For additional data, discussion, and references, see Fig. S9 and its extended figure caption provided in the SI of the revised manuscript. We emphasize that Mn donation to DCPIP can occur even under high light intensity and pronounced photoinhibition typically proceeds after the electron donor is consumed or in the case where no electron donor is present (see, e.g., D. J. Blubaugh, G. M. Cheniae, 1990, Kinetics of photoinhibition in hydroxylamine extracted photosystem II membranes: relevance to photoactivation and sites of electron donation. *Biochemistry* 29, 5109-5118).

In summary, continuous electron donation in the presence of low MnCl₂ concentration for more than 5 min disfavors that termination of the electron donation and DCPIP reduction results from photoinhibitory damage (the high and low [MnCl₂] use the same light intensity). Previously published results addressing the low-concentration MnCl₂ effect, the inverse relation between continuous slow electron donation (slow indicated by low slope of DCPIP curves) continuing for more than 5 min and the fast electron transfer (high slope) resulting in Mn oxide formation terminated within 1 min—this syndrome of observations is explainable by assuming that the formed Mn oxide nanoparticles inhibit electron donation via Tyrz once they have reached a certain size.

In support we have added the Figures S7, S8, and S9, and we say explicitly (on pg. 6): "Notably, at low MnCl₂ concentration DCPIP reduction continued at undiminished rate for a clearly longer illumination time (> 5 min, Figs. 4 and S7), suggesting that photoinhibitory damage does not rapidly terminate electron transfer in the Mn-depleted PSII."

In addition, Mn oxide particles can produce harmful hydrogen peroxide under UV light. The Schott KG5 filter used in this work for illumination might be insufficient.

<3b> Thanks for noting and motivating text modifications to allay concerns on this point. We have checked and the KG5 filter transmission at 300 nm is 1% and 10⁻⁴ at 280 nm cutting off the UV-C range, but not the UV-B and UV-A range and thus it indeed might be insufficient. However, also the UV-B radiation is completely eliminated and the UV-A light strongly weakened because of both, the emission characteristics of the used halogen lamp and the cut-off of the light guide. Since a direct light effect on Mn oxides inducing a photochemical process requires high light intensities (even more so for disordered Mn oxides with short exciton diffusion ranges and lifetimes), we can safely exclude peroxide formation that results from direct light excitation of Mn oxides. In the Materials and Methods section of the revised manuscript we now say: "The combination of light source, KG5 filter, light guide, and cuvette material resulted in an effective spectral range of about 400 nm to 750 nm (limits correspond to the 10% level) effectively excluding UV irradiation, thereby minimizing potential interferences due to peroxide formation resulting from direct Mn-oxide excitation, and sample heating due to infrared light (thermal radiation)." (pg. 14/15).

The authors find that the Mn-oxide cluster size are limited to around 100 Mn ions, which may correspond to a nanoparticle that could fit within the Photosystem II cavity. The Mn/PSII ratios of table S1 were obtained by an estimation that all the Photosystem II centers are active during the measurement time. If inhibitory effects are present, the calculated cluster size would be much bigger and could not fit anymore into the Photosystem II cavity possibly compromising the authors statement that these clusters are biomineralized and fixed by Photosystem II.

<3c> Thank-you. We have clarified the writing and addressed potential photoinhibitory effects now as discussed in the response to comments above (<3a> and <3b>.)

Minor comments to the manuscript are:

-The use of buffer and pH in this work is quite confusing. All experiments use Mes as buffer with pH adjusted to 7, 7.5 and 8 although Mes has no buffering capacities from pH 7.5. While all pH values are used to present optical measurements, XAS measurements (Figure 5, S10, S11) use only Photosystem II particles prepared at pH 8. No comments in the manuscript are made on the choice and influence of pH and buffer to the measurements and, particularly, to the production of Mn oxide nanoparticles.

<3d> Thank-you for this point, which we agree needed clarification. The pH was routinely monitored prior to and after the illumination assays in the actual cuvette and found to be stable within ± 0.2 pH units within a time period of at least 20 min, as now stated in the Methods section in the paragraph on 'Optical absorption spectroscopy and illumination procedures' (pg. 14).

(What accounts for the absence of major pH changes even under conditions where the buffering capacity is not optimal? The total number of protons released in the course of Mn oxide formation is expected to be low because of the low concentration of PSII in the illumination solution (ca. 0.1 μM) and further diminished by proton uptake at the PSII acceptor side, coupled to DCPIP and PPBQ reduction. For 100 Mn ions bound to PSII, we estimate a $\text{Mn}^{3+/4+}$ concentration of 10 μM and a net release of protons corresponding to 10-40 μM . This means that a tiny buffer capacity will be sufficient, likely still provided by 25 mM MES buffer supported by the non-negligible buffering capacity of the PSII membrane particles.)

Various experiments on Mn-depleted PSII are presented in main manuscript and the SI that have been performed at pH 6.5, pH 7, pH 7.5, pH 8.0 and pH 8.5. The motivation of the pH choice for the two main experiments is explained in the following:

DCPIP reduction: In Figs. 3 and 4, we present DCPIP experiments for illumination of PSII membrane particles at pH 7, because proton-coupled DCPIP reduction is significantly slowed down at more alkaline pH, as shown in Fig. S5A, hindering a meaningful comparison with the electron donation rate of the intact PSII. Spectra for DCPIP reduction at pH 8.5 are shown in Fig. S11.

XAS experiments: The XAS experiments shown in Fig. 5 as well as the TXRF results of Table 1, were performed using PPBQ as electron acceptor and illumination at pH 8.5, because we expect the Mn oxides to be more stable at higher pH during centrifugation and XAS

sample preparation. However, as an additional control, X-ray data at the BESSY synchrotron was collected also for Mn-depleted PSII illuminated and kept at pH 7. In the revised manuscripts, Figure S14 compares the EXAFS spectra of Mn oxides formed at pH 8.5 and pH 7, suggesting that their structure is the same (see also EXAFS simulations in Table S2), thereby supporting that the same birnessite-type Mn oxides are formed irrespective of pH.

We have pursued this study over several years and consider it to be especially comprehensive, with a high number of controls and supporting experiments. The above choice of collected and presented data as well controls at different pH is, in our opinion, reasonable and well suited to support our conclusions. In the revised manuscript, we state more clearly the pH used in the respective experiments. In this context, we apologize sincerely for an error in the originally submitted manuscript, where the stated pH values of the XAS and TXRF experiment were incorrect and now have been corrected.

-In Figure S9, the efficiency of light induced Mn oxide production is compared for different exogenous electron acceptors. The slow hydrophilic, but most electro-positive, electron acceptor ferricyanide only produces low amounts of Mn oxides which is in favor of the production of biomineralized nanoparticles at the donor side that necessitates high electron turnover. However, to ensure that the reduced hydrophobic electron acceptors are not implicated in Mn oxide nanoparticle production, a control using a mixture of a hydrophobic electron acceptor and ferricyanide is missing.

<3f> The use of DCPIP as a relatively hydrophilic electron acceptor supports Mn-oxide formation well, i.e. similar to the hydrophobic acceptor PPBQ (Fig. S9), evidencing that a hydrophobic acceptor is not essential for Mn-oxide formation. The different amounts of Mn-oxide formed with the different acceptors likely reflect their particular properties, especially the (unwanted) ability to react with the already formed Mn oxides, but most likely not their direct involvement in oxide formation. We added a respective statement to the Fig. S9 caption; a further study of the acceptor effects is not expected to provide significant insight in the oxide formation itself.

-The spectra of Figure S4 don't have much of interest. If at all, it would be more instructive to see the influence of 240 μ M MnCl₂.

<3g> We agree that Figure S4 does not have much interest and thus have removed it from the SI of the revised manuscript.

-The experiment of Figure S5 is a control. However, the concentration of DCPIP (45 μ M) is not the same as used for other experiments.

<3h> We apologize for this error that occurred during writing of the manuscript. The used concentration was 60 μ M, as now correctly stated also in the figure caption (now Fig. S4).

----- *In summary ..* -----

In summary, the actual state of the manuscript still lacks clear experimental evidence that the Mn oxide nanoparticles are biomineralized and fixed by Photosystem II which is the essential conclusion of this manuscript and which distinguishes it from other propositions found in literature. If such additional experimental evidence is provided by the authors, it would be worth publishing in Nature communication.

<3i> We appreciate this thoughtful, albeit skeptical comment. It has prompted us to clarify our results, especially regarding the separation techniques (see above), and materially respond by way of presenting additional work. Additional experiments have now been provided as detailed further above. In response to the summarizing statement of the reviewer, we want to emphasize the following:

I) We feel that our central result is the formation of Mn oxide nanoparticles by photosystem II, which is of clear relevance for when discussion the three evolutionary questions listed on pg. 13, irrespective of whether today PSII binds Mn oxide particles directly to its protein matrix or not. As detailed in the background text of the SI, all previous studies on the topic could conclude only circumstantially that Mn oxide might be formed, but never provided direct verification by structural characterization. This achievement clearly distinguishes our study from previous ones.

II) However, we also see that specifically the evolutionary path suggested in Figure 6 of the manuscript can gain support by evidence that the Mn oxide nanoparticles are bound to PSII. We believe that sufficient evidence for this conjecture is provided by our study, as explained in <3a> and <3b>, and we emphasize that in the course of the revision, informative additional experiments on this point have been pursued. The additional data is now presented in form of several SI figures (Figs. S7-S9, S15, S16) and referred to in the main article.

REVIEWERS' COMMENTS

Reviewer #1 (Remarks to the Author):

The authors have responded adequately (both in their rebuttal letter and through modifications applied to the ms) to my points of critique and my suggestions. I also went through their responses to items raised by the other reviewers and find that also their grievances were satisfactorily dealt with. I therefore now consider this ms as suitable for publication in Nature Communications.

Reviewer #3 (Remarks to the Author):

The revised version of the manuscript addresses all my concerns in a clear and concise writing and the responses are well argued. The only point for which I am not fully convinced, is that the same photoinhibition mechanisms take place at low and high Mn²⁺ concentrations, since high turnover of electron transfer might induce additional inhibitory effects. However, this is not an important question for the manuscript, since the authors provide now strong evidence against my main concern, that the MnO₂ nanoparticles may not be bound to PSII. They provide new analysis of the data supporting competitive inhibition by the nanoparticles at the donor side of PSII and new experiments with solubilized PSII, which show that PS II most likely binds these particles. From these experiments, it cannot be totally excluded that PEG interacts with unbound MnO₂ nanoparticles and I think that the analysis of solubilized PSII obtained by native gel electrophoresis could exclude this possibility. In summary, I am favorable for the publication of the manuscript in its current state and I suggest to the authors to try the native gel electrophoresis experiment if they want to present direct evidence.

Response to Reviewers comments

Reviewer #1:

The authors have responded adequately (both in their rebuttal letter and through modifications applied to the ms) to my points of critique and my suggestions. I also went through their responses to items raised by the other reviewers and find that also their grievances were satisfactorily dealt with. I therefore now consider this ms as suitable for publication in Nature Communications.

<<We thank the reviewer for the positive recommendation!

Reviewer #3:

The revised version of the manuscript addresses all my concerns in a clear and concise writing and the responses are well argued. The only point for which I am not fully convinced, is that the same photoinhibition mechanisms take place at low and high Mn²⁺ concentrations, since high turnover of electron transfer might induce additional inhibitory effects. However, this is not an important question for the manuscript, since the authors provide now strong evidence against my main concern, that the MnO₂ nanoparticles may not be bound to PSII.

<<We agree that there exists the possibility of different contributions of effects to photoinhibition at low and high Mn²⁺ concentrations. This is certainly of interest for future investigations, but, as the Reviewer also states, not an important question for the present study (e.g., because photoinhibition is not an important issue here) so that we postpone respective experiments.

They provide new analysis of the data supporting competitive inhibition by the nanoparticles at the donor side of PSII and new experiments with solubilized PSII, which show that PS II most likely binds these particles. From these experiments, it cannot be totally excluded that PEG interacts with unbound MnO₂ nanoparticles and I think that the analysis of solubilized PSII obtained by native gel electrophoresis could exclude this possibility.

<<We agree that native gels may contribute to answering the question whether PEG interacts with MnO₂ particles. However, in agreement with the Reviewer's recommendation for publication of the ms in the present state (below), we feel that such experiments are not essential here (and may even not be conclusive) to shed further light on the Mn-oxide formation by PSII and will be included in a forthcoming publication.

In summary, I am favorable for the publication of the manuscript in its current state and I suggest to the authors to try the native gel electrophoresis experiment if they want to present direct evidence.

<<We thank the Reviewer for the positive recommendation and will include gel experiments in an upcoming manuscript (see comments above).